# Unveiling the Terahertz Nano-Fingerprint Spectrum of Single Artificial Metallic Resonator

**DOI:** 10.3390/s24185866

**Published:** 2024-09-10

**Authors:** Xingxing Xu, Fu Tang, Xiaoqiuyan Zhang, Shenggang Liu

**Affiliations:** 1Terahertz Research Center, School of Electronic Science and Engineering, University of Electronic Science and Technology of China, Chengdu 611731, China; 202011022904@std.uestc.edu.cn (X.X.); 202112022427@std.uestc.edu.cn (F.T.); liusg@uestc.edu.cn (S.L.); 2Key Laboratory of Terahertz Technology, Ministry of Education, Chengdu 611731, China

**Keywords:** subwavelength, THz metasurface, THz-TDS s-SNOM, nanoscale spectrum

## Abstract

As artificially engineered subwavelength periodic structures, terahertz (THz) metasurface devices exhibit an equivalent dielectric constant and dispersion relation akin to those of natural materials with specific THz absorption peaks, describable using the Lorentz model. Traditional verification methods typically involve testing structural arrays using reflected and transmitted optical paths. However, directly detecting the dielectric constant of individual units has remained a significant challenge. In this study, we employed a THz time-domain spectrometer-based scattering-type scanning near-field optical microscope (THz-TDS s-SNOM) to investigate the near-field nanoscale spectrum and resonant mode distribution of a single-metal double-gap split-ring resonator (DSRR) and rectangular antenna. The findings reveal that they exhibit a dispersion relation similar to that of natural materials in specific polarization directions, indicating that units of THz metasurface can be analogous to those of molecular structures in materials. This microscopic analysis of the dispersion relation of artificial structures offers new insights into the working mechanisms of THz metasurfaces.

## 1. Introduction

THz metasurface devices have long been utilized in electromagnetic emission [1,2,3], control [4,5,6,7], imaging [8] and biochemical sensing [9,10,11,12], owing to their distinctive electromagnetic properties, with metallic resonant structures being among the most prevalent applications. The interaction between these structures and electromagnetic waves is predominantly explained using the LC circuit resonance model [13,14] and the equivalent dielectric constant model [15,16]. However, due to the sub-wavelength nature of metasurface elements, their interaction efficiency with THz waves is inherently low, necessitating the use of large-scale arrays to observe their influences on electromagnetic waves effectively. Through traditional THz far-field transmission and reflection experiments, researchers have demonstrated that the resonant absorption properties of arrays of artificial metal structures exhibit similarities to those of natural materials with specific absorption peaks in the THz frequency band, such as lactose [17], lead, and vermilion [18], the permittivity of which can be described by the Lorentz model. Nonetheless, artificial structures differ from natural materials in that their equivalent dielectric constant can be freely tuned by adjusting structural parameters, thereby offering broader application scenarios.

In THz metasurface devices, the electromagnetic characteristics of each unit can be significantly altered by micron-level parameter changes, thereby almost directly determining the electromagnetic properties of the entire array. The detection and analysis of individual elements are crucial for enhancing our understanding of the interaction mechanisms between metasurfaces and electromagnetic waves. Traditional far-field imaging methods are constrained by spatial resolution limitations, necessitating the diffraction limit to be broken to directly detect the equivalent dielectric constant or surface field distribution of a single metasurface element. The advent of THz near-field imaging technology has increased the imaging and spectral resolution of THz waves to the micron and even nanometer scale [19,20], enabling the direct measurement of nanoscale phenomena that have experienced rapid advancements in recent years, such as nanoresonators, surface plasmons [21], and phonon-polaritons [22]. Among these technologies, photoconductive probe, aperture, and scattering THz near-field imaging systems have reached considerable maturity in recent years. The fundamental principle of the first two methods involves positioning the detector at a distance of within a few or tens of microns from the sample surface for near-field detection [23,24]. Recent advancements have seen research teams utilizing these methods to achieve micron-level imaging and spectral formation of single-symmetric bimetallic antennas [25] and asymmetric resonant rings [26]. The scattering near-field system leverages an atomic force microscope (AFM) probe to scatter near-field information from the sample surface, thereby achieving imaging resolutions comparable to those of AFM [27,28,29,30]. Paul Dean’s team has recently employed this system to observe the near-field surface resonance modes of single-rectangular-metallic antenna and ring structure [31]. Building on these studies, we designed a DSRR structure and a simple rectangular antenna, then employed THz-TDS s-SNOM to simultaneously obtain the THz broadband near-field intensity and phase information from its surface. This approach enabled the two-dimensional imaging of the surface resonance mode and the dispersion analysis of the equivalent dielectric constant in the resonance region. Additionally, by incrementally altering the position of one of the gaps, we observed notable differences between far-field and near-field detection.

## 2. Materials and Methods

### 2.1. Experimental Setup

The THz-TDS s-SNOM used in the experiment is a commercial product from Neaspec GmbH, and is composed of AFM and THz-TDS. The schematic diagram of the experimental optical path is illustrated in Figure 1. A collimated P-polarized THz pulse is obliquely incident and converged at the tip of the AFM probe, which operates in tapping mode, via an off-axis parabolic mirror with a focal length of 16 mm and an incident angle of approximately 52 degrees. In this configuration, the local field between the probe and the sample is modulated at the probe’s tapping frequency and scattered into free space. Both the reflected and scattered signals are captured by a photoconductive antenna (PCA) receiver. When demodulating high-order near-field signals, which is carried out to extract relatively pure THz near-field information, the process is typically executed at each time delay point. In conventional far-field time-domain scanning, the signal at each time delay point remains relatively stable. However, in near-field systems, the signal at each time delay point encapsulates modulated scattered near-field information, rendering it inherently unstable. These weak near-field components can be extracted from the signal received by the PCA by utilizing a phase-locked demodulation system, which demodulates the higher-order harmonics of the probe’s oscillation frequency. By scanning the delay line, the time-domain signal of the near-field can be incrementally reconstructed. The probe used in this setup is a 25PtIr500B-H50 type, customized by RMN Inc, the cantilever length of which exceeds 500 μm and the tip shank of which is about 80 μm, featuring a resonant frequency of 15.5 kHz and an amplitude of approximately 150 nm. To mitigate the influence of water vapor on the THz spectrum, the system operates in an environment with humidity maintained below 10%.

### 2.2. Material Preparation

Figure 2a,b present scanning electron microscope (SEM) images of the sample and a schematic diagram of a single DSRR structure, respectively. The parameters of the designed DSRR are as follows: *p* = 70 μm, L = 40 μm, w = 6 μm, and s = 2.5 μm. The horizontal distance between the center of the upper gap and the center of the structure, defined as the asymmetric coefficient, ac, is another critical parameter. In addition, 25 metallic rectangular antennas of different lengths were fabricated, each with a uniform width of 3 μm and lengths ranging from 38 μm to 86 μm in 2 μm increments. To minimize coupling effects and mutual interference between adjacent antennas, the inter-antenna spacing was designed to exceed 300 μm. In the experiment, these structures were fabricated from gold using a photolithography process, achieving a thickness of approximately 100 nm. The substrate material was high-resistance silicon (HR-Si) with a resistivity exceeding 10,000 Ω·cm and a thickness of 1 mm.

### 2.3. Methods of Detection

To achieve such high imaging resolution in the s-SNOM system, the most crucial aspect is the extraction of pure near-field signals amidst substantial background noise. Enhancing near-field scattering efficiency is typically accomplished by utilizing a metallic tip within the THz frequency band. The metallic tip significantly amplifies the local electric field at the probe tip due to its pronounced lightning rod effect, while simultaneously leveraging the mirror dipole interaction between the probe tip and the sample surface to scatter the surface electric field. Consequently, the spatial extent of the near-field electric field is primarily determined by the curvature radius of the probe tip. The quantization of the scattered signal can be effectively modeled via the polarization and radiation processes of the point dipole formed between the probe tip and the sample, commonly referred to as the point dipole model of the near field [32]. In this model, the probe tip is approximated as a sphere with radius *r* . As the probe oscillates with a certain frequency, Ω, and amplitude, *A*, the minimum distance between the probe tip and the sample surface is *z*. Following the probe’s oscillation, the distance, *d*, between the center of the probe tip and the sample surface can be expressed as follows:$$d=z+r+A\left[1+cos\left(2\pi \Omega t\right)\right]$$

At the same time, the polarizability, α, of the probe tip can be expressed as follows:$$\alpha =4\pi r^{3}\left(\varepsilon_{t}-1\right)/\left(\varepsilon_{t}+2\right)$$

In the aforementioned equation, $\varepsilon_{t}$ represents the dielectric constant of the probe material. When a P-polarized THz signal impinges upon the probe tip, the effective polarizability of the probe sample dipole is given by the following:$$\alpha_{eff}=\alpha /\left(1-\alpha \beta /\left(16\pi d^{3}\right)\right)$$
where β is related to the sample reflection coefficient, which is calculated via $\beta =\left(\varepsilon_{s}-1\right)/$$\left(\varepsilon_{s}+1\right)$, including $\varepsilon_{s}$ for the dielectric constant of the sample. Combined with the reflection coefficient, γ, of the sample surface and the incident electric field intensity, $E_{i}$, the final scattered signal, $E_{s}$, of the probe can be expressed as follows:$$E_{s}\propto \alpha_{eff}{\left(1+\gamma \right)}^{2}E_{i}$$

The above equation represents the modulation process of the probe on the near-field signal, where demodulation involves determining the Fourier expansion coefficients of the effective polarizability. As the probe retracts from the sample surface, the local field between the probe and the sample undergoes significant variation, decreasing exponentially with increasing distance from the sample. In contrast, the reflected background signal decreases almost linearly with increasing distance due to the large spot size. Therefore, theoretically, higher-order demodulation yields purer near-field signal extraction. However, due to the characteristics of Fourier expansion in exponentially decaying signals, higher-order near-field signals are inherently weaker, resulting in a lower signal-to-noise ratio (SNR) in practical experiments. Thus, it is crucial to select an appropriate order based on the specific experimental conditions.

### 2.4. Data Processing

Due to the normalization of the THz pulse being influenced by the time delay and the unknown flatness of the sample surface, a substrate region within tens of micrometers near the structure is typically selected when acquiring the reference near-field signal. This approach ensures a minimal delay difference between the sample and the reference signal. During the time-domain near-field signal scan, each delay point is integrated for 200 ms to achieve a sufficient SNR for the 2nd-order signal. While the 3rd-order near-field signal can also be observed under these conditions, the reference signal generally exhibits a relatively lower SNR. The normalization uses the following method:$$\mathrm{Norm}S_{2}\left(\omega \right)=\left|\frac{S_{2,sample}\left(\omega \right)}{S_{2,ref}\left(\omega \right)}\right|$$
(1)$$\mathrm{Norm}P_{2}\left(\omega \right)=P_{2,sample}\left(\omega \right)-P_{2,ref}\left(\omega \right)$$
where $S_{2,sample}$ and $P_{2,sample}$ are the 2nd-order near-field amplitude and phase obtained in the surface of our DSRR, respectively. Also, $S_{2,ref}$ and $P_{2,ref}$ are the same kinds of data obtained in the surface of substrate, which is HR-Si. In phase processing, it may be necessary to subtract the baseline. In addition, the far-field reflection spectrum of periodic structures in the following figures should be normalized to the reflection spectrum of the pure substrate, which is also HR-Si.

## 3. Results

Initially, AFM topography and third-order near-field time-domain peak imaging were performed on a single symmetric DSRR with an asymmetric coefficient of ac = 0 μm, which landed in the center area of the array. The results are presented in Figure 2c. Due to the scanning range approaching the maximum imaging region of the AFM (57 μm × 66 μm), some distortions are observed at the edges of the topograph and near-field images. Nevertheless, these distortions do not impact the analysis of the near-field signal strength. It is evident that the near-field signal from gold is significantly higher than that from high-resistance silicon, while the near-field signal strength from gold remains nearly uniform. To ascertain the distribution of the resonance mode, it is imperative to collect the time-domain signal at each point and image the single-frequency point. To determine the actual resonant frequency of the designed DSRR, the far-field reflection spectrum of the periodic structure was measured. For comparative purposes, array structures with varying asymmetric coefficients (ac = 0, 2, 4, 6, and 8 μm) were fabricated on the same silicon substrate. Each array comprised 100 elements (10 × 10) spanning an area of 700 μm × 700 μm, with an inter-array spacing exceeding 1.3 mm (see Figure A1), ensuring that the THz spot illuminated only one array at a time. In the experiment, the far-field reflection and near-field tests share the same optical path, allowing the far-field measurements to reliably predict the near-field resonance frequency. The far-field reflection spectra for different asymmetric coefficients are depicted in Figure 2d. These results indicate that all absorption peaks are centered around 0.73 THz (λ≈ 6p), with larger asymmetric coefficients corresponding to lower absorption peak frequencies. This observation suggests that lower frequencies correlate with the resonance of a longer arm, implying that the absorption peak is predominantly influenced by the long arm of the structure.

According to the antenna effect of the THz probe [33,34] and the finite dipole model theory [32], the probe in the THz near-field system primarily couples and scatters the external component of the sample surface. The $E_{z}$ contour plot and surface current distribution of the asymmetric DSRR at the resonant frequency, obtained through CST simulation, are presented in Figure 3b. We can see that the surface current forms a closed loop, indicating that the resonant mode corresponds to a magnetic dipole mode. Additionally, the electric field resonance is predominantly concentrated near the vicinity of the split gap, highlighting the localization of the electromagnetic field in that region. Figure 3a illustrates a schematic diagram of the optical path used in our actual test. The THz wave is obliquely incident from the left side, with the probe positioned on a metal arm near the opening, specifically at the red dot indicated in Figure 3b. For an asymmetric coefficient of ac = 0 μm, the second-order time-domain near-field signal and corresponding normalized second-order near-field intensity and phase spectra with a time window of 2–10 ps measured at the red dot are shown in Figure 3c. Approximately 4 ps behind the main peak (indicated by the gray dashed line box), a peak misalignment between the substrate and the DSRR begins to manifest, leading to the subsequent spectral differences. We can see that the intensity and phase spectra align closely with the real and imaginary parts of the dielectric constant, as described by the typical Lorentz model. In 2019, Haewook Han’s team reported near-field observations of α-lactose and β-lactose, calculating the equivalent permittivity of these lactose molecules using the probe sample line dipole model and HDPE as references, achieving a good fit with the Lorentz model [35]. Similar structures have been tested in the infrared near-field for organic materials such as polystyrene (PS) and polymethyl-methacrylate (PMMA) [36]. The near-field model reveals that the strength of the near-field signal is positively correlated with the real part of the dielectric constant, and the phase of the near-field signal is positively correlated with the imaginary part of the dielectric constant. Consequently, the equivalent dielectric constant extracted from the near-field information at the electric field resonance locations on the metasurface is consistent with the Lorentz model. Using the Lorentz model formula,
(2)$$\varepsilon \left(\omega \right)=\varepsilon_{\infty }+\frac{\omega_{p}^{2}}{\omega_{0}^{2}-\omega^{2}-i\gamma \omega }$$
a preliminary fit can be applied to the intensity and phase of the second-order near-field normalization using the Lorentz model. In the equation, $\omega_{0}$ and γ represent the resonant frequency and damping parameters, respectively. The fitting results are depicted by the dashed line in Figure 3c, with the resonant frequency, according to the fitting coefficients, these also being 0.73 THz. To further elucidate the resonant size-sensitive properties of the metasurface elements, we varied the asymmetric coefficients of the DSRR and conducted near-field experiments on individual structures. The second-order normalized near-field phase spectra for different asymmetric coefficients are shown in Figure 3d. The testing positions remained consistent, revealing significant changes in the resonant frequency of a single resonant ring. Although there are minor discrepancies between the resonant frequencies and the positions of the far-field absorption peaks, the trend is consistent with that observed in the far-field measurements.

As a typical rectangular antenna within a metallic metasurface structure, although the resonance phenomena are less pronounced compared to those of resonant rings, it still enables efficient tuning of the resonant frequency. Additionally, an isolated metallic rectangular antenna typically functions as a half-wave antenna, with its resonant frequency primarily dependent on its physical length and the dielectric constant of the substrate material. The resonant wavelength exhibits direct proportionality to the antenna’s length, facilitating the analysis of the experimental results. The optical image of some metallic rectangular antennas is shown in Figure 4a, and Figure 4b illustrates the antenna’s structural diagram (top) and the corresponding simulation contour diagram at the resonant frequency (bottom), where the electric field in the simulation is an out-of-plane component, consistent with the characteristic resonant mode distribution of a half-wave antenna. Figure 4e presents the optical path diagram for both the simulation and the actual experiment. To conduct a more detailed analysis of the antenna’s resonance, we initially compared the normalized second-order near-field time-domain signals of metallic antennas with varying lengths, as shown in Figure 4c. Given that the water vapor content in the experimental environment was below 3%, the near-field time-domain signals in the pure substrate exhibited only the primary and secondary reflections resulting from surface wave propagation on the probe [37]. When the near-field signal was collected at the edge of the metal antenna (indicated by the red dot in Figure 4b), a clear observation could be made of the electric field oscillation occurring subsequent to the main peak in the near-field signal. This oscillation persisted until the reflected signal from the probe cantilever’s end reached the probe tip and was scattered, corresponding to the reflection time in the time domain. For longer rectangular antennas, it was observed that the period of electromagnetic oscillation increased correspondingly, directly confirming the resonant behavior of THz waves on metallic antennas through time-domain analysis.

Based on the time-domain signal, it is possible to predict the corresponding characteristics in the frequency domain. Figure 4d presents the FFT results of the time-domain signal shown in Figure 4c. In this analysis, a signal within the time window from 1 ps to 6.5 ps—spanning from the main peak to the primary reflection—was selected. However, due to the prolonged duration of resonance in the metal antenna, this method may have resulted in a reduced measured near-field resonance Q-factor. The choice of different time windows for signal extraction introduced minor variations in the measured resonance frequency (refer to Figure A2b), but it did not affect the overall linearity of the resonant spectrum or the trend of resonant frequency changes for metal antennas of varying lengths. In addition, the resonant spectra could still be accurately modeled using the Lorentzian profile. We observed that the trend of resonant frequency corresponding to metal antennas of different lengths aligned with our predictions. Additionally, the relationship between the resonant wavelength and the lengths of all 25 antennas is plotted as a scatter plot in Figure 4f, with a linear fit applied. The slope of the fitted line was determined to be 5.708, allowing us to approximate that the THz propagation velocity in a metal with a high-resistance silicon substrate, whose reflective index is 3.4, is approximately c/2.854. The dielectric properties of the engineered metasurface elements were further validated through near-field spectral testing of the metallic rectangular antenna.

To obtain the resonant field distribution on the surface of the DSRR, time-domain scanning was performed along the metal edge of the DSRR with a scanning interval of 1 μm. By performing FFT for each point and normalizing it to the substrate, a phase contour diagram for a single frequency point, as shown in Figure 5c, was obtained (the phase of substrate is defined as 0 rad). To eliminate inter-structure interactions within the array, this DSRR was positioned in an isolated location with no other structures within a 400 μm radius. The corresponding frequency gradually increased from bottom to top, with 0.73 THz being the resonant frequency of the DSRR. It can be observed that the electric field of the resonance is primarily concentrated on both sides of the opening and gradually weakens as it moves away from the opening. The resonance disappears at the 2.5 μm opening, fully demonstrating the spatial resolution of the near-field scattering. Figure 5a,b show the near-field phase spectra obtained via scanning along the red and green dashed lines of the DSRR in Figure 5c, respectively. It can be intuitively observed that there is essentially no resonant electric field in the central region of the opening and the unopened edges. For the metallic rectangular antenna, we also conducted near-field spectrum tests at different locations, the results of which are basically consistent with the simulation results. The details are shown in Figure A2a.

## 4. Discussion

Although we observed the field distribution and nanoscale spectrum on the surface of the metasurface element, when we pay attention to the simulation results (see Figure 3b) and the experimental results (see Figure 5c), it can be found that a phase difference between the two sides of the opening was not detected, and the same result can also be observed in metallic rectangular antennas (see Figure 4b and Figure A2a), which may be caused by the disturbance of the surface electric field by the probe. The metal probe’s lightning rod effect causes a large accumulation of charge at the tip [38], forming a strong local electric field. Consequently, when the probe is near the surface of the structure, some coupling between the local field and the resonant field on the surface of the structure inevitably occurs, affecting the intensity and phase information of the scattered field. This phenomenon was also noted by other research teams in previous studies [39].

Given that the resonant mode of the DSRR corresponds to a magnetic dipole, the entire ring effectively constitutes a resonant wavelength. Consequently, the measured results should approximate the resonant frequency of an 80 μm antenna in a half-wave antenna configuration. As observed in Figure 3, the resonant frequency of the DSRR is approximately 0.72 THz, which closely aligns with the resonant frequency of the 76 μm rectangular antenna. Considering the inherent uncertainties in structural fabrication and experimental measurement, it is reasonable to conclude that the two structures exhibit a strong correspondence in their resonant behaviors.

Additionally, it is noteworthy that as we reduce the asymmetry coefficient, the absorption peak in the far-field reflection spectrum continuously diminishes (see Figure 2d). In the case of structural symmetry, the absorption peak is almost imperceptible. Even when the sample is rotated by 90 degrees, no significant far-field absorption peak is observed (as shown in Figure 6). However, near-field tests reveal a strong resonance near 0.73 THz, indicative of a typical radiation interference cancellation phenomenon. Previous studies by Longqing Cong’s team demonstrated that symmetric structures produce the bound states in the continuum (BIC) phenomenon under a normally incident Y-polarized electric field due to the radiation interference of two symmetric metal arms [40]. Although our experiment involves oblique incidence, the two metal arms remain symmetric relative to the incident THz wave, causing interference cancellation. When the asymmetry coefficient increases and the structure loses its symmetry, the interference cancellation condition is no longer met, and the structure’s resonance is reflected in the far-field absorption peak.

## 5. Conclusions

In conclusion, we employed THz-TDS s-SNOM to conduct nanoscale spectrum testing of a single metallic DSRR and to achieve sub-micron imaging of its resonant mode. By extracting the near-field spectrum and normalizing it to that of high-resistance silicon substrate, we found that the intensity and phase conformed to the typical Lorentz model, aligning closely with results obtained for natural resonant materials. At the same time, the near-field spectrum of rectangular metal antennas of different lengths was observed, and the propagation speed of THz in metal with silicon substrate was estimated roughly. Additionally, we simulated both resonant and non-resonant frequencies on the surface of the structure with a sub-micron resolution, directly observing that the resonant field was concentrated on both sides of the opening and decayed towards the metal center. For the symmetric DSRR, the persistence of near-field resonance indicates that the disappearance of the absorption peak in the far-field reflection is due to interference cancellation of structural radiation. These testing methodologies can be broadly applied to various metallic metasurface structures. Our findings offer novel insights into the working mechanisms of metasurface units at a microscopic level and are anticipated to advance the development of THz emission, regulation, and sensing technologies.

## Figures and Tables

**Figure 1 sensors-24-05866-f001:**
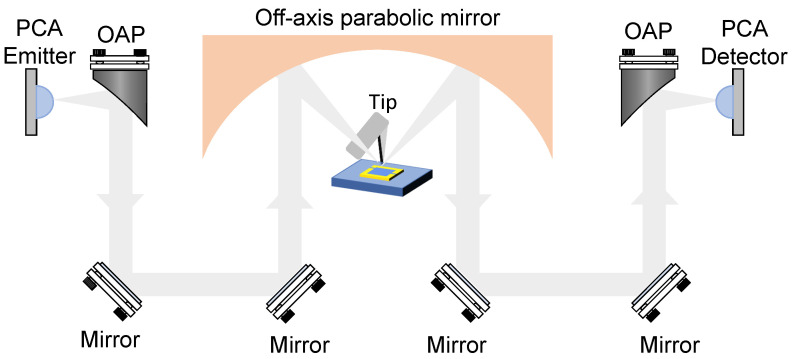
Schematic diagram of the THz-TDS s-SNOM.

**Figure 2 sensors-24-05866-f002:**
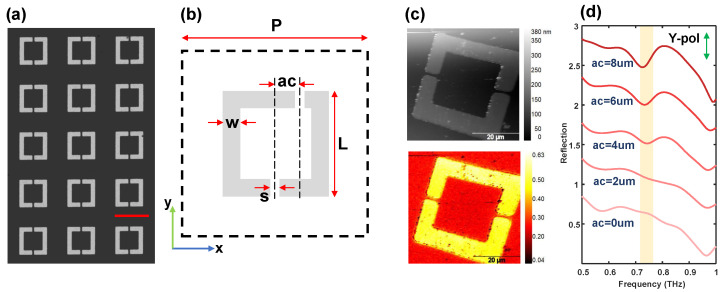
(**a**) SEM image of a symmetric DSRR array; the scale bar is 50 μm. (**b**) A schematic diagram of DSRR, in which ac is the asymmetric coefficient. (**c**) Topography (top) and 3rd-order near-field imaging of the time-domain peak (bottom). (**d**) The normalized far-field reflection spectrum of DSRR arrays with different asymmetric coefficients. We can see that the absorption peak increases and undergoes a redshift as the ac increases.

**Figure 3 sensors-24-05866-f003:**
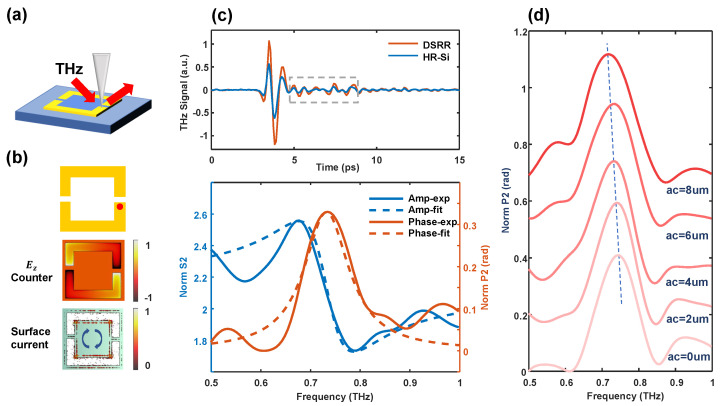
(**a**) A schematic diagram of our near-field experiment. (**b**) A simple diagram of the symmetric DSRR; the red dot indicates where the tip scatters the near-field signal (top). The contour of the z component of the electric field (center) and surface current (bottom) as ac = 2
μm. (**c**) The top picture shows the 2nd-order near-field time-domain signal measured at the base and at the red dot, and the bottom shows the 2nd-order near-field spectrum of the DSRR normalized to the HR-Si substrate, where the real and dashed lines correspond to the experimental and Lorentz fitting results, respectively. (**d**) The normalized 2nd-order near-field phase spectrum measured at the red dot with the increasing ac from the bottom up.

**Figure 4 sensors-24-05866-f004:**
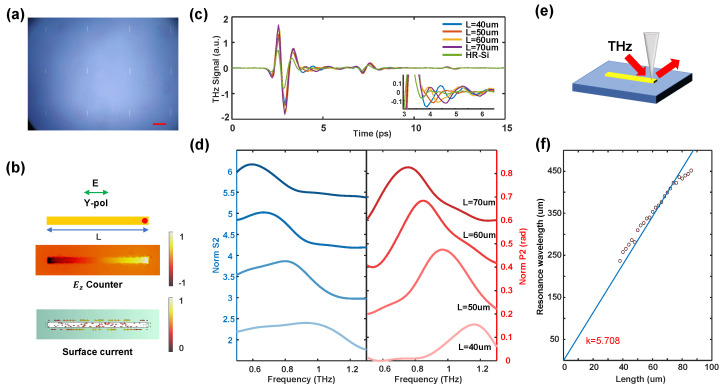
(**a**) An optical image of metallic rectangular antennas; the scale bar is 50 μm. (**b**) A simple diagram of a rectangular antenna; the red dot indicates where the tip scatters the near-field signal (top). A contour picture of the z component of the electric field (center) and surface current (bottom) as L = 60 μm. (**c**) The 2nd-order near-field time domain signal measured at the base and at the red dot when L = 40, 50, 60 and 70 μm. The embedded picture is an enlarged graph of the signal. (**d**) The corresponding 2nd-order near-field amplitude and phase spectrum normalized to those of the HR-Si substrate. (**e**) A schematic diagram of our near-field experiment. (**f**) The experimentally measured resonant wavelengths of rectangular antennas with different lengths (black circles) and corresponding linear fitting results (blue solid lines).

**Figure 5 sensors-24-05866-f005:**
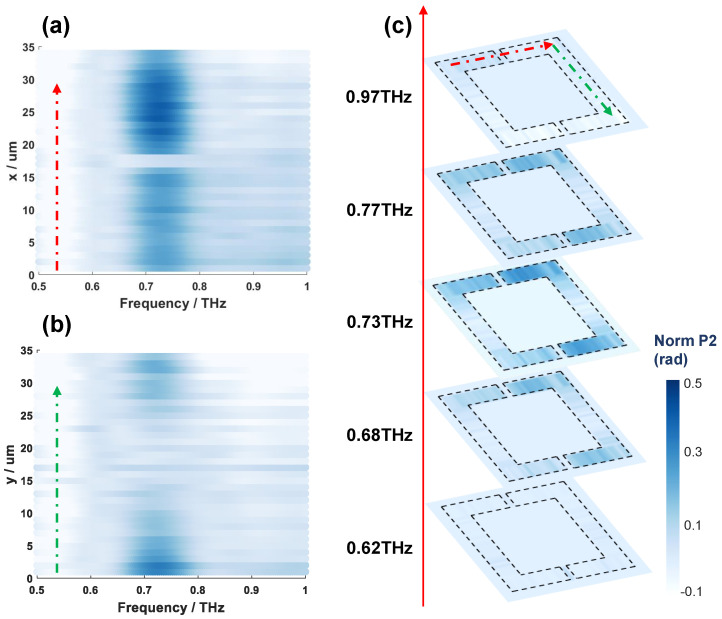
(**a**) The line scanning the normalized 2nd-order near-field phase spectrum of the opening side, where the scanning direction and range correspond to the red dashed line in (**c**). (**b**) The same as (**a**), except the scanning direction and range correspond to the green dashed line in (**c**). (**c**) Normalized 2nd-order near-field phase contour diagram with different frequencies (the resonant frequency of this DSRR is 0.73 THz).

**Figure 6 sensors-24-05866-f006:**
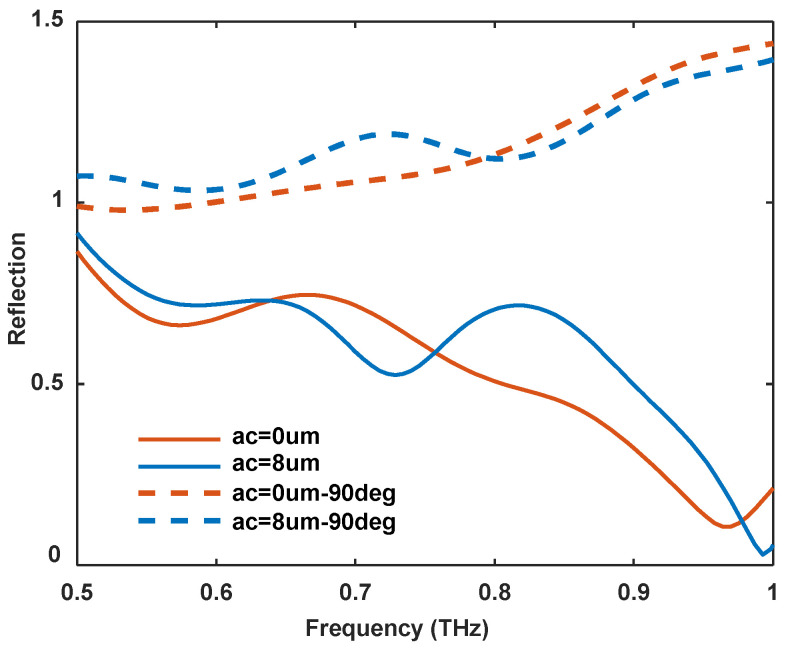
Normalized far-field reflection spectrum of symmetric DSRR array at two angles perpendicular to each other. The solid lines are the same as those in Figure 2d, and the dashed lines are measured by rotating the sample by 90 degrees. When ac is 0 μm, the array has no apparent absorption in both conditions. Since the reference signal comes from the HR-Si substrate, it is reasonable for the normalized reflection of the periodic structures to be greater than 1.

## Data Availability

The data presented in this study are available at the links mentioned in the text or on request from the corresponding author.

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
