# Peer review of "Unveiling the Terahertz Nano-Fingerprint Spectrum of Single Artificial Metallic Resonator"

_sensors, 2024, doi:10.3390/s24185866_

Round 1

Reviewer 1 Report

Comments and Suggestions for Authors

The paper shows how a terahertz time-domain spectrometer is used to investigate the near-field nanoscale spectrum and the resonant mode distribution of a metallic double-gap split ring resonator. The authors combine the TDS with the atomic force microscope in order to measure both far and near fields. So, the electromagnetic fields near a metasurface units is studied at a sub-micron level. The investigations conducted show that the intensity and phase of the resonance conformed to the typical Lorentz model. These measurements performed with sub-micron resolution, demonstrate that the resonant field was concentrated on both sides of the slots and decayed towards the metal center. The experimental set up presents some novelty and able to be used in the next studies. The paper falls into a scope of the journal. However, the paper cannot be published in the presented view.

  1. The first phrase in Abstract does not contain any information specific to the paper under review. It can be excluded.
  2. In Discussion, the absorption peak in the far-field reflection spectrum is considered with the reference to Figure 3(c). But this figure shows the 2nd order near-field time domain signal and near-field spectrum. 
  3. In Experimental setup section, more detailed description how the near-field time-domain signals are obtained would be worthy.
  4. In the line 92 the explanation of AFM abbreviation is excessive.

Reviewer 2 Report

Comments and Suggestions for Authors

Authors employed a scattering-type scanning near-field optical microscope based on a terahertz time-domain spectrometer (THz-TDS s-SNOM) to investigate the near-field nanoscale spectrum and resonant mode distribution of a single metal double-gap split ring resonator. The authors present the results of simulation and experiment, which are quite attractive. Because most of the research focuses on the whole list of structures, the authors here study individual structures.

1. Please explain "3rd order near field time domain ..." and "2nd order...", The reviewer is confused here.

2. At the bottom of formula 1, there is a mistake in the writing of P and S, so check it carefully.

3. The reflectance spectral characteristics of periodic structures are given in the study of reflectance spectra, not the individual structures studied. Is the author's research and title right? Is the topic appropriate? It requires careful consideration by the author.

4. In Fig.5, Why the refelction is larger than 1?

5. Authors are advised to describe when individual structures are studied, when array structures are studied, and what advantages and disadvantages they have.

6. In the introduction, some background information needs to be added, such as the recommended references below.

(1)Opto-Electron Adv 5, 210062 (2022). doi: 10.29026/oea.2022.210062.

(2)Optics and Lasers in Engineering, 2024, 177,108128

(3)Opto-Electron Adv 6, 220012 (2023). doi: 10.29026/oea.2023.220012.

(4)Opto-Electron Sci 2, 220026 (2023). doi: 10.29026/oes.2023.220026.

(5)Opto-Electron Adv 6, 230153 (2023). doi: 10.29026/oea.2023.230153

(6)Opto-Electron Sci 2, 230025 (2023). doi: 10.29026/oes.2023.230025

Round 2

Reviewer 1 Report

Comments and Suggestions for Authors

The paper shows how a terahertz time-domain spectrometer is used to investigate the near-field nanoscale spectrum and the resonant mode distribution of a metallic double-gap split ring resonator. The authors combine the TDS with the atomic force microscope in order to measure both far and near fields. So, the electromagnetic fields near a metasurface units is studied at a sub-micron level. The investigations conducted show that the intensity and phase of the resonance conformed to the typical Lorentz model. These measurements performed with sub-micron resolution, demonstrate that the resonant field was concentrated on both sides of the slots and decayed towards the metal center. The experimental set up presents some novelty and able to be used in the next studies. The paper falls into a scope of the journal.

Few corrections are necessary before publication.

  1. Exclude repeated designations (AFM) AFM in line 122.
  2. Through the text the word-combination “terahertz THz” is excessive.  After this combination is mentioned for the first time, the later quite enough to write either “terahertz” or  “THz”.

Author Response

Comments 1: Exclude repeated designations (AFM) AFM in line 122.

Response 1: Thank you for pointing this out. I agree with this comment, and the modification can be found in page 4, line 115.

Comments 2: Through the text the word-combination “terahertz THz” is excessive. After this combination is mentioned the first time, the later quite enough to write either “terahertz” or “THz”.

Response 1: Thank you for pointing this out. I agree with this comment. The first appearance of “terahertz (THz)” can be found in abstract section, line 1, and all the words “terahertz” in the following article have been changed to "THz". These changes can be found in line 3, 6, 10, 12, 15, 21, 23, 25, 30, 37, 38, 42, 53, 60, 61, 62, 69, 79, 106, 129, 138, 139, 146, 199, 215, 266, 270, 276, 284.

The texts in blue color in the former revised manuscript, which are original contents in the first submission, have been deleted, and those in red color in the former revised manuscript, which are modifications in the first review process, have been changed to black color.